# Systematic Review of the Occurrence and Antimicrobial Resistance Profile of Foodborne Pathogens from *Enterobacteriaceae* in Wild Ungulates Within the European Countries

**DOI:** 10.3390/pathogens13121046

**Published:** 2024-11-28

**Authors:** Răzvan-Tudor Pătrînjan, Adriana Morar, Alexandra Ban-Cucerzan, Sebastian Alexandru Popa, Mirela Imre, Doru Morar, Kálmán Imre

**Affiliations:** Faculty of Veterinary Medicine, University of Life Sciences “King Mihai I” from Timișoara, 300645 Timișoara, Romania; adrianamorar@usvt.ro (A.M.); alexandracucerzan@usvt.ro (A.B.-C.); sebastian.popa@usvt.ro (S.A.P.); dorumorar@usvt.ro (D.M.); kalmanimre@usvt.ro (K.I.)

**Keywords:** wild ungulates, *Salmonella*, *E. coli*, *Yersinia*, antimicrobial-resistance

## Abstract

Game meat is derived from non-domesticated, free-ranging wild animals and plays an important role in human nutrition, but it is recognized as a source of food-borne and drug-resistant pathogens impacting food safety. The present review aimed to provide a comprehensive analysis of the frequency of isolation and antimicrobial resistance (AMR) profiles of major foodborne pathogens from the *Enterobacteriaceae*, including *Salmonella*, *Escherichia*, and *Yersinia* genera, in wild ungulates, across Europe in the 21st century. A systematic search was conducted via the Google Scholar database using the PRISMA guidelines. In this regard, the content of a total of 52 relevant scientific publications from both European Union (n = 10) and non-European Union countries (n = 3) was processed, highlighting the main scientific achievements and indicating knowledge gaps and future perspectives. The studies highlighted that *Salmonella* spp. was the most commonly encountered pathogen, and significant AMR levels were noticed for the isolated strains, especially against penicillin (32.8%) and amoxicillin (32.1%). This review underscores the importance of monitoring the presence of food-borne pathogens and their AMR in wildlife as important public health and food safety concerns.

## 1. Introduction

Game meat is derived from non-domesticated, free-ranging wild animals and birds that are either legally hunted for personal consumption or raised, slaughtered, and commercially processed for food [1]. Game meat has played a significant role in human nutrition, and game hunting has remained an essential activity in many parts of the world, including European countries [2]. Approximately seven million hunters are registered in Europe and are pivotal in the primary production sector of game meat. European countries with the highest absolute numbers of registered hunters include France (approximately 1.3 million), Spain, the United Kingdom (UK), and Italy [2]. In rural areas within mainland Europe, hunters are regarded as primary producers of game meat, with an important contribution to the development of local economies, supporting thus sustainable meat production [1,2].

Game meat production in Europe decreased almost nine times in 2020, reaching a total production of about 13.5 thousand tons, compared to the first year analyzed, 2005, when there was a production of 129 thousand tons. According to the FAO (Food and Agriculture Organization of the United Nations) data from 2017, Germany led the rankings in terms of hunting meat production, with a total quantity of 58,400 tons, followed by Sweden with 16,062 tons, and Poland with 8103 tons [1].

While the foodborne pathogens affecting livestock and game animals in Europe are largely similar (e.g., *Salmonella* spp. and *Trichinella* spp.), the absence of a standardized surveillance program for game meat is a significant issue. This lack of uniform safety protocols and policies across European countries further complicates the management of these issues [2]. Numerous pathogens associated with game animals pose significant risks to the public and animal health [3]. These pathogens can be transmitted through various routes, and only a subset are food-borne or can be transmitted via droplets produced during the processing of infected animal carcasses [2].

Wildlife plays a critical role as a reservoir for zoonosis, especially pathogenic enteric bacteria [1]. Game meat is likely to be more contaminated with enteric micro-organisms than meat from domestic animals due to several highly variable factors during the harvesting (e.g., hunting practices or the conditions under which game carcasses are dressed). *Enterobacteriaceae* are the indicator bacteria for the microbiological quality of food and the hygiene status of a production process. Additionally, the food contaminated by *Enterobacteriaceae* poses a microbiological risk for consumers [4].

The *Enterobacteriaceae* family, including species such as *Salmonella*, *E. coli*, *Proteus*, and *Klebsiella*, presents a significant challenge in raw and processed meat products worldwide [1,3]. These bacteria are predominant in food poisoning cases linked to various meat products. Given the unique challenges posed by game meat, it is crucial to include it in safety measures to effectively address its specific contamination risks [5]. Salmonellosis is an enteric infectious disease that significantly threatens public health. In 2022, it was the second most frequently reported foodborne zoonosis in the European Union, with 65,208 cases of human illness. There were also 1014 foodborne outbreaks resulting in 6632 cases of illness, 1406 hospitalizations, and 8 deaths [6]. Additionally, yersiniosis accounted for 7919 reported cases and 636 hospitalizations during the same period. It is important to note that game meat was not identified as a source in these reports.

Wildlife is usually not exposed to clinically used antimicrobial agents but can acquire antimicrobial resistance (AMR) through contact with humans, domesticated animals, and environments [7]. The spread of AMR bacteria in wildlife must be viewed as a major concern with serious implications for human and animal health. AMR is a growing global concern in the field of food safety and public health [8]. Game meat, sourced from wild animals such as deer, wild boar, and game birds, has gained popularity in recent years. However, limited research has been conducted to assess the presence and prevalence of AMR bacteria in game meat. Understanding the prevalence of AMR bacteria in game meat is crucial, as it can help identify potential risks to human and animal health. Antimicrobials are necessary agents to fight diseases in humans, animals, plants, and crops. Despite this, their use is complicated by the development of AMR. One of the major causes of this natural process is the overuse of antimicrobials and their inappropriate administration (e.g., wrong category of antimicrobials, inadequate dose, and reduced duration of therapy), and this results in a high quantity of failed treatments against pathogens and in increasing mortality. AMR has been detected particularly among commensal gut bacteria with patterns that vary across species, locations, and times [4,5,6]. The increasing number of antimicrobial-resistant *Enterobacteriaceae* both in veterinary and human medicine, the dissemination of these bacteria in several environments, and their possible repercussions on human health are causing concern. [5]

Taking these considerations into account, this review aims to provide a comprehensive summary of the frequency of isolation and AMR profiles of major foodborne pathogens from *Enterobacteriaceae*, specifically focusing on the genera *Salmonella*, *Escherichia*, and *Yersinia* in game animals across European countries in the 21st century.

## 2. Methods

This study followed the Preferred Reporting Items for Systematic Reviews (PRISMA) guidelines. The process for the systematic review is detailed in Figure 1. Initially, a thorough exploration was conducted to identify peer-reviewed scientific publications concerning the prevalence and AMR of *Enterobacteriaceae* members in wild ungulates across a major database: Google Scholar (last searched on July 2024). Additionally, the European Food Safety Authority was consulted (last searched on July 2024). The search was limited to studies conducted in Europe between 2001 and 2024. This geographical restriction was applied to ensure relevance and contextual specificity, considering potential variations in game meat consumption practices, environmental factors, and regulatory frameworks across different regions. The outcomes targeted in this review included the prevalence of foodborne pathogens in wild ungulates and their AMR profiles.

The occurrence of specific pathogens, such as *E. coli*, *Salmonella* spp., and *Yersinia* spp., in tested samples from various wild ungulates and the patterns of AMR exhibited by these pathogens. The searching methodology employed specific keywords, including “game meat”, “*Enterobacteriaceae*” and “antimicrobial resistance”, supplemented by additional terms like “roe deer (*Capreolus capreolus*)”, “red deer (*Cervus elaphus*)”, “wild boar (*Sus scrofa*)”, “chamois (*Rupicapra rupicapra*)”, “moose (*Alces alces*)”, “fallow deer (*Dama dama*)”, “mouflon (*Ovis gmelini*)”, “*E. coli*”, “*Salmonella* spp.”, and “*Yersinia* spp.”, strategically combined to ensure inclusivity. Other food-borne pathogens of *Enterobacteriaceae*, such as *Cronobacter* and *Shigella,* were excluded due to the limited availability of comprehensive data on these pathogens within mainland Europe and game meat. *Cronobacter* spp., for instance, while recognized for its severe implications in neonatal infections, particularly through contaminated powdered infant formula, remains less frequently reported in foodborne disease surveillance compared to more prevalent pathogens like *Salmonella* or pathogenic strains of *E. coli* [9]. Similarly, *Shigella* infections, though significant, are often overshadowed by other more commonly reported enteric pathogens, and the variability in reporting practices can lead to an incomplete epidemiological picture [10]. In an initial search, a total of 437 articles were identified. The eligibility of the articles was based on the availability of information regarding the prevalence of the targeted genera *Salmonella*, *Escherichia,* and *Yersinia*, as well as data on AMR.

All selected studies were published in peer-reviewed journals, organizational websites, books, and dissertations, and were exclusively in the English language. The initial screening phase involved assessing the titles of the articles, with exclusions made for irrelevant studies. This included duplicates, studies focused on other samples’ origin, those concerning other animal species such as domestic animals (n = 33), other wild species (n = 41), or wild birds (n = 5), and studies related to other bacterial species (n = 23). In the subsequent selection phase, the abstracts of the remaining studies were independently and thoroughly reviewed to determine their relevance to the study’s objectives.

Three independent reviewers extracted data from each included report. The reviewers worked independently, and any discrepancies were resolved through discussion or consultation with a 4th reviewer if necessary. No attempts were made to contact study authors for data confirmation, as all required information was available from the reports. Data extraction was conducted manually without the use of automation tools.

Information was systematically extracted from each article, including the author, year of publication, country of the study, wild species investigated, number of samples tested, number of positive samples, number of isolates obtained, and data on AMR. Initially, a total of 437 manuscripts were identified through the Google Scholar database. Of these, 21 publications were excluded as their titles were either completely irrelevant or they were duplicates. The abstracts of the remaining 404 articles were then reviewed, resulting in the exclusion of 209 articles that did not align with the predefined criteria for this review. This exclusion was based on the following reasons: irrelevance to the scope of the review (n = 113), non-English language publications (n = 21), and lack of clear identification of *Enterobacteriaceae* (n = 75).

Consequently, 185 studies remained for full-text examination, of which 13 could not be retrieved. Of the 172 articles fully reviewed, 33 were excluded as they only concerned domestic animals, 46 for concerning only other wild species, 14 for not reporting the number of positive samples, 2 for not reporting the total number of samples, 23 for discussing other bacterial species (e.g., *Campylobacter*, *Listeria*), and 2 for a non-European country (e.g., Egypt and Namibia). Ultimately, 52 manuscripts met the inclusion criteria and were incorporated into this review. We prioritized results that reported the most recent and relevant data, focusing on studies that provided the highest methodological quality and comprehensiveness in their findings.

## 3. Results and Discussions

### 3.1. Isolation Frequency of Salmonella in Hunted Game Animals

*Salmonella* spp. is a Gram-negative, flagellated, facultative anaerobic bacteria that belongs to the *Enterobacteriaceae*, and more than 2500 serotypes are known [11,12]. Salmonellosis is an enteric infectious disease that poses a hazard for meat safety. With 65,208 cases of human illness, 1014 foodborne outbreaks causing 6632 cases of illness, 1406 hospitalizations, and 8 deaths, it was the second-most-often reported foodborne zoonosis in the European Union in 2022 [13]. *Salmonella* spp. has been identified as a high-priority concern in ensuring the safety of wild boar meat and is recognized as a significant biological hazard in wild animals [14,15]. Despite this, the contribution of game meat to the epidemiology of human salmonellosis remains unexplored.

Regarding the prevalence of *Salmonella* spp. in game animals, scientific publications conducted in Europe and covering 12 countries, including Norway, Sweden, Germany, Czech Republic, Switzerland, Slovenia, Italy, Portugal, Spain, Romania, Serbia, and Greece, were reviewed. The published results about the prevalence of *Salmonella* in wild ungulates are summarized in Table 1. It is noteworthy that the recorded prevalence values of *Salmonella* spp. in game animals are highly variable, ranging from 0% in Norway, Germany, and Switzerland to 47.7% in Slovenia, as can be observed in Figure 2.

The most frequently isolated *Salmonella* serotypes obtained from game meat samples were *Salmonella* Salamae, accounting for a total of 83 isolates, followed by *Salmonella* Diarizonae with 73 isolates and *Salmonella* Enterica with 40 isolates. These findings highlight a notable prevalence of *S. salamae* and *S. diarizonae* in the sampled populations, suggesting that these serotypes may be more commonly associated with wild game meat. This distribution of serotypes provides valuable insights into the patterns of *Salmonella* contamination in wild game and raises important considerations for food safety monitoring and control practices [39].

The data compiled from various studies reveals significant variability in the prevalence of *Salmonella* spp. among wild animal populations, particularly in wild boars, across different countries. For instance, Vieira-Pinto et al. [27] reported a prevalence of 22.1% in wild boars, underscoring their potential role as carriers and spreaders of *Salmonella* spp. This finding was supported by the results published by Razzuoli et al. [39], who found a 12.45% prevalence, yet lower than the 19.3% in Murcia, Spain [26], and 35% in Latium, Italy [35]. Similarly, Rîmbu et al. [31] identified a 10.7% prevalence, whereas Hulánková et al. [33] reported a very low prevalence of 0.4% in the Czech Republic, illustrating significant geographical variability.

Further studies, such as that by Bonardi et al. [37], found a 2% and 10.2% prevalence in both carcasses and mesenteric lymph nodes, respectively, while Bassi et al. [19], who reported a 17% seroprevalence in Switzerland, revealed substantial differences even within relatively close regions. These results are mirrored by the findings of Cilia et al. [38], who documented a 4.18% prevalence, comparable to studies in Spain [25] and Sweden [29]. Additionally, Razzuoli et al. [39] found 540 out of 4335 samples were positive for *Salmonella* spp., indicating a notable presence of the pathogen. Petrović et al. [20] reported an overall prevalence of 1.6% in Vojvodina hunting grounds, with some areas reaching up to 33.3%, suggesting localized spikes in prevalence. Meanwhile, Siddi et al. [40] observed an overall prevalence of 4.5%, and Floris et al. [41] found no *Salmonella* spp. in muscle and liver samples tested by PCR. Altissimi et al. [6] reported an overall prevalence of 1.36% in a comprehensive study from 2018 to 2023.

In contrast to wild boars, other wild ungulates such as deer, chamois, moose, and ibex exhibit lower or no prevalence of *Salmonella* spp. [22]. Obwegeser et al. [18] and Lillehaug et al. [16] reported no detection of *Salmonella* spp. in their samples, while Díaz-Sánchez et al. [24] found an overall sample-level prevalence of 0.8%, with 1.2% in wild boars and 0.3% in red deer. These findings suggest that these species present a lower risk of *Salmonella* transmission compared to wild boars.

Comparing prevalence rates across different regions and species, Ortega et al. [26] reported a 19.3% overall *Salmonella* seroprevalence in Spain, higher than 1.5% in Portugal [28], 7.2% in Italy [36], and 11.3% in Northeast Spain [23]. However, this rate is lower than the 47.7% in Slovenia [21] and 30.7% in Campania, Italy [34], but higher than the 4.3% in Greece [32] and 12.4% in Switzerland [17].

The most frequently identified serotypes of *Salmonella* were *S. enterica* subsp. Salamae, with 83 isolates [25,35,39,40], followed by *S. diarizonae* with 73 isolates [25,29,35,37,39], *S. enteritidis* with 37 isolates [17,20,37,39], and *S. typhimurium* with 30 isolates [6,20,27,35,39].

In addition, other serotypes with significant public health implications were identified: *S. paratyphi* (1) and *S. newport* (10) [35,38,39]. *S. paratyphi* is responsible for paratyphoid fever, which poses substantial health risks due to its transmission through contaminated food and water [42], while *S. newport* is particularly concerning due to its frequent association with multidrug resistance (MDR), complicating treatment efforts and leading to severe, widespread outbreaks [43]. Both serotypes underscore the importance of vigilant monitoring and stringent control measures to mitigate their impact on public health.

The reviewed studies confirm that wild boar populations serve as significant reservoirs for *Salmonella* spp., posing potential risks to human and animal health [6,7,8,9,10,11,12,13,14,15,16,17,18,19,20,21,22,23,24,25,26,27,28,29,30,31,32,33,34,35,36,37,38,39,40,41,42,43]. The variability in prevalence rates underscores the importance of localized studies to assess specific risk factors and implement targeted public health interventions. While other wild animals such as deer, chamois, and ibex present a lower risk, the increasing wild boar populations and their interaction with human activities elevate the potential for zoonotic transmission. Further research should focus on identifying the specific *Salmonella* serotypes circulating within wild ungulate populations to better understand the epidemiology and develop effective control measures. Given the public health implications, it is essential to monitor the impact of the growing wild boar populations and the consumption of their meat, implement rigorous food safety practices, and educate the public on the risks associated with handling and consuming wild game meat. This comprehensive approach will aid in mitigating the risks posed by *Salmonella* spp. and safeguarding public health.

### 3.2. Distribution of the Pathogenic Escherichia coli Strains in Hunted Game Animals

Generic *E. coli* is often a harmless component of the normal microflora in humans and other animals. Nevertheless, acquiring virulence genes through various mechanisms has endowed certain *E. coli* strains with different types of pathogenicity. Numerous enteropathogenic groups of *E. coli* have been identified as causes of various gastrointestinal infections. Six principal pathotypes of *E. coli* have been distinguished: enteropathogenic *E. coli* (EPEC), enterotoxigenic *E. coli* (ETEC), enteroinvasive *E. coli* (EIEC), diffusely adhering *E. coli* (DAEC), enteroaggregative *E. coli* (EAEC), and enterohemorrhagic *E. coli* (EHEC) [44,45]. According to the European Food Safety Authority (EFSA), the European Centre for Disease Prevention and Control (ECDC), and the One Health 2020 Zoonoses Report, STEC (Shiga toxin-producing *Escherichia coli*) infections rank as the fourth most common zoonotic disease, following campylobacteriosis, salmonellosis, and yersiniosis [46]. Currently, the pathogenicity of STEC is categorized by serotype, with the top five being O26, O103, O111, O145, and O157, which were previously the most frequently detected serotypes among patients with hemolytic uraemic syndrome (HUS) [47]. Public health authorities mainly focus on O157 STEC infections due to their high pathogenicity. However, non-O157 STEC serogroups, including O26, O103, O111, O121, and O145, cause twice as many human infections [48]. This serotype is a component of the gut microbiota in various animal species, with ruminants, particularly cattle, identified as a major reservoir [49,50]. Wild boars have been previously identified as carriers of *E. coli* O15 [51,52] and other STEC strains that pose potential pathogenic risks to humans [17,24,53]. It is important to note that the reported prevalence rates of *E. coli* in game animals exhibit significant variability, as shown in Figure 3. The published results regarding the prevalence of *E. coli* in game ungulates are summarized in Table 2.

In a study by Lillehaug et al. [16], 104 isolates of potentially pathogenic serovars of *E. coli* were identified among the 207 pooled samples examined. The serovar *E. coli* O103 was detected in 41% of the pooled samples, whereas serovars O26 and O145 were found less frequently. Notably, serovars O111 and O157 were not observed in any of the samples. In contrast, in a study conducted in Switzerland [18], an analysis of 239 fecal samples from wild ungulates revealed that 53.1% were positive for *E. coli*. Similarly, in a study by Sánchez et al. [55], STEC strains were detected in 58 (23.9%) of the animals sampled. The prevalence of STEC was found to be 24.7% (51 out of 206) in red deer, 5% (1/20) in roe deer, 33.3% (2/6) in fallow deer, and 36.4% (4/11) in mouflon. Additionally, two different STEC strains were identified in seven of the animals. Díaz-Sánchez et al. [24] isolated STEC from deer and wild boar in the carcass and fecal samples. The overall prevalence of STEC in fecal samples was 21.6% (124/574), while in carcass samples it was 21.3% (125/585). Non-STEC O157 strains were isolated in 34% (89/264) of deer fecal samples, 4% (11/301) of wild boar fecal samples, 7% (19/271) of deer carcass samples, and 4% (12/310) of wild boar carcass samples. Similarly, in a study by Mora et al. [52], STEC strains were recovered from 52.5% of the roe deer (94/179) and 8.4% of the wild boars (22/262). Lauzi et al. [62] found similar data in the analyzed deer feces, with a prevalence of STEC strains of 19.9%. Out of 536 fecal samples tested from wild boars tested by Plaza-Rodríguez et al. [63], 37 yielded STEC (6.9%). Considering the species, STEC was recovered from 37% of the red deer samples (37/101) and 14% from wild boar (8/56) samples. These results followed the European trend, with a higher STEC rate identified in cervids [57].

The findings obtained by Szczerba-Turek et al. [60] demonstrated that red deer and roe deer serve as potential carriers of non-O-157 STEC isolates, which may pose pathogenic risks to humans. STEC strains were identified in 21.65% and 24.63% of rectal swabs from red deer and roe deer, respectively. This finding is particularly significant given Europe’s steadily increasing red deer population and the direct and indirect interactions between wildlife and humans, domestic animals, water bodies, and the broader environment. The results further corroborate that wildlife, including wild ungulates, can act as reservoirs and disseminators of STEC within the environment.

In a study by Díaz-Sánchez et al. [64], *E. coli* O157 was specifically detected and isolated from deer fecal samples at four of the thirty-three hunting estates sampled, corresponding to a prevalence of 12% at the estate level and 1.5% (4/264) at the sample level. Likewise, Navarro-Gonzalez et al. [65] reported that *E. coli* O157:H7 was found in 4/117 wild boars, yielding a prevalence of 3.41%, and in 2/160 Iberian ibexes, with a prevalence of 1.25%. Similar findings were obtained in the Czech Republic [33], where *E. coli* O157 was detected in 2/242 fecal samples (0.8%).

In Serbia [54], during the 2013–2014 hunting season, fecal samples collected from roe deer, deer, and wild boars were sent to the bacteriology laboratory for isolation. Out of 105 fecal samples, *E. coli* was isolated from 100 samples, indicating a high prevalence rate of 95.23%. Similarly, in Portugal, during the same investigation period, 67 fecal samples were collected from wild ungulates, including wild boar, red deer, and roe deer. *E. coli* was detected in 96% of the samples (64) [28]. A high prevalence of *E. coli* (83.78%) was detected in Poland by Wasyl et al. [59], where, out of 660 fecal samples from wild boar, roe deer, red deer, and fallow deer, 553 were positive for *E. coli*. In Tuscany, Bertelloni et al., [61] reported that 175 *E. coli* pure cultures were isolated from 200 tested animals (87.5%). These findings highlight a comparably high prevalence of *E. coli* in wild, ungulate populations across different regions of Europe.

Based on the data, it is noteworthy that various species of wild ungulates across different regions of Europe serve as significant reservoirs for STEC and other pathogenic *E. coli* strains. The prevalence rates of STEC and non-O157 STEC isolates vary considerably among species and regions, with red deer, roe deer, and wild boars frequently identified as carriers. Studies consistently report notable prevalence rates, such as 21.6% in deer fecal samples and 6.9% in wild boar fecal samples [64], indicating a widespread distribution of these pathogens in wildlife. Additionally, specific pathogenic serovars like *E. coli* O103, O26, and O145 are detected more frequently, while O157 remains relatively rare in some studies but present in others at varying levels [16,65]. These findings underscore the role of wild ungulates as important vectors in the environmental dissemination of STEC, posing potential risks to human and animal health due to their increasing interactions with humans and domestic animals.

### 3.3. Yersinia spp. in Game Animals

The genus *Yersinia*, belonging to the bacterial family *Enterobacteriaceae*, comprises 28 distinct species. Among these, three species, *Y. pestis*, *Y. pseudotuberculosis*, and *Y. enterocolitica*, have been identified as pathogenic to humans. Notably, *Y. enterocolitica* is prevalent across a variety of food sources, animal reservoirs, and environmental niches and comprises both pathogenic and non-pathogenic strains [66].

In Europe, *Y. enterocolitica* strains (serotype O:3 and serotype O:9) are often associated with clinical cases in humans [67]. Swine is an important reservoir of these bioserotypes, and they usually carry the agent asymptomatically in the tonsils [68].

In the EU, yersiniosis is classified as a notifiable zoonotic disease, mandating reporting to authorities and inclusion in the European Food Safety Authority’s (EFSA) annual report [69]. Most cases of human yersiniosis reported in Europe are caused by *Y. enterocolitica,* and only a few have been attributed to *Y. pseudotuberculosis*. According to the European Centre for Disease Prevention and Control, a total of 7663 cases of yersiniosis were reported across the European Union (including the UK) in 2019. Of these cases, 100 were attributed to *Y. pseudotuberculosis*, while 7563 were caused by *Y. enterocolitica*. The highest notification rates were observed in member states located in northeastern Europe [70]. It is important to note that none of these infections were caused by consumption of game meat. In wildlife, European authors reported a prevalence between 58.2% and 2.5% in Spain [71] and Italy [72], respectively. The overall prevalence for each country can be observed in Figure 4.

The published data on the prevalence of *Yersinia* spp. in game meat are succinctly summarized in Table 3.

From the data analyzed across 6 European countries, *Y. enterocolitica* was the most frequently isolated species. This prevalence underscores the widespread distribution of *Y. enterocolitica* within the region, emphasizing its potential significance as a public health concern across diverse geographic areas in Europe [69].

The contamination of game meat with *Y. enterocolitica* also remains insufficiently investigated [74]. Pathogenic *Y. enterocolitica* was detected on the surface of 38.3% of raw game meat samples in Bavaria, Germany.

In Switzerland, Fredriksson-Ahomaa et al. [73] reported a 44% detection rate of enteropathogenic *Yersinia* in the tonsils of 153 wild boars using real-time PCR. Specifically, *Y. enterocolitica* was detected in 35% of the animals, while *Y. pseudotuberculosis* was found in 20%. Notably, both species were simultaneously detected in 10% of the sampled wild boars.

In Italy [78], a total of 28 *Yersinia* spp. were isolated from 18 out of 251 animals (7.2%): ten wild boar (15.4%), four red deer (7.1%), three roe deer (4.9%) and one chamois (1.5%). Six of these were identified as *Y. enterocolitica* species; these six isolates were retrieved from one chamois, one roe deer, one red deer, and three wild boars. Similarly, in Spain, Arrausi-Subiza et al. [71] reported that antibodies against *Y. enterocolitica* and *Y. pseudotuberculosis* were detected in 52.5% of the tested animals. Using PCR, *Y. enterocolitica* was identified in 33.3% of the wild boars, while *Y. pseudotuberculosis* was detected in 25% of the tonsil samples. Correspondingly, the study conducted by Sannö et al. [29] provides valuable insights into the prevalence of *Yersinia* species in wild boars. A comprehensive analysis of 319 samples from 88 wild boars was performed using PCR, including 175 tonsil samples, 88 fecal samples, and 56 ILN samples. The results demonstrated a significant presence of pathogenic *Yersinia* species among the sampled population. Specifically, 20.5% (18/88) of the wild boars tested positive for *Y. enterocolitica*, while 19.3% (17/88) were positive for *Y. pseudotuberculosis*. Four individuals tested positive for both *Y. enterocolitica* and *Y. pseudotuberculosis*.

In a study conducted by Von Altrock et al. [75], the tonsils of 111 wild boars hunted in Lower Saxony, Germany, were investigated. A total of 17.1% of the wild boar tonsils were positive for *Y. enterocolitica*, while two boars (1.8%) carried isolates identified to be *Y. frederiksenii*. Likewise, bacteriological examination of 302 rectal swabs from 151 wild boars in Poland resulted in the isolation of 40 *Y. enterocolitica* strains [76].

Laboratory examination of 336 swabs collected from 56 wild ungulates carcasses in Poland revealed 52 *Y. enterocolitica* strains. These were identified in 12/20 (60%) of roe deer carcasses, 7/16 (43.8%) of red deer carcasses, and 11/20 (55%) of wild boar carcasses. The relatively high degree of carcass contamination with *Y. enterocolitica* is of concern due to the growing popularity of game meat among consumers [77].

The findings of Syczyło et al. [69] provide compelling evidence of the widespread presence of *Y. enterocolitica* among game animals in Poland. The study revealed that *Y. enterocolitica* isolates were detected in the rectal swabs of 21.7% (186/857) of the tested animals, the prevalence of *Y. enterocolitica* infections (110/434) was highest in wild boars, where 25.3% of the examined animals were infected. In comparison, only 21.6% of red deer (63/291), 9.4% of roe deer (11/117), and 13.3% of fallow deer (2/15) were infected. In a study conducted by Sannö et al. [30] in Sweden, 31.0% (28/90) of wild boars tested positive for *Y. enterocolitica*, while 22.0% (20/90) were positive for *Y. pseudotuberculosis*.

In the study conducted by Bonardi et al. [37], *Y. enterocolitica* was not isolated from carcasses or MLNs. However, 3/49 carcasses were contaminated with bacteria of the genus *Yersinia*. Specifically, one strain each of *Y. frederiksenii*, *Y. bercovieri*, and *Y. aldovae* was detected from these carcasses. Notably, these species are not recognized as causative agents of human yersiniosis.

In contrast, Cilia et al. [38] reported that 71 *Yersinia* isolates were obtained from rectal swabs of wild boars, accounting for 24.7% of the samples analyzed. Of these, 54 isolates (18.8%) were biochemically identified as *Y. enterocolitica*. The remaining 17 isolates were classified as *Y. frederiksenii* or *Y. intermedia*. Similarly, Siddi et al. [40] reported an overall prevalence of *Y. enterocolitica* of 30.3% (20/66) in wild boars. Among the *Y. enterocolitica*-positive animals, 10% (2/20) tested positive in both colon content and carcass surface samples, and an additional 10% (2/20) were positive in both colon content and MLNs. Furthermore, 5% (1/20) of the animals were positive for *Y. enterocolitica* exclusively in carcass surface samples. Specifically, *Y. enterocolitica* was identified in 27.3% (18/66) of colon content samples, 4.5% (3/66) of MLN samples, and 6.1% (3/49) of carcass surface samples.

In a study conducted by Floris et al. [41], *Y. enterocolitica* in 101 liver samples was analyzed, detecting a total of eight strains (7.9%), distributed as follows: five from wild boars, two from chamois, and one from deer. The prevalence of *Y. enterocolitica* in this study was approximately 8%, which is higher than the 2.5% prevalence reported for Liguria [72] but lower than the prevalence documented in Tuscany [38].

This comprehensive review of studies across various European countries highlights the significant presence and varying prevalence rates of *Y. enterocolitica* and other *Yersinia* species in wild ungulates, particularly wild boars. Despite the observed variations in prevalence rates, the overall findings indicate a widespread distribution of these pathogens, with substantial evidence identifying wild boars as significant reservoirs. Under these circumstances, it is imperative to characterize the strains within wild boar populations to ascertain their serotype, biotype, and, most critically, their pathogenic potential. The reviewed studies highlight the importance of ongoing monitoring and detailed characterization of *Yersinia* strains in wild game populations to enhance our understanding of their pathogenic potential and to inform and develop effective public health strategies.

### 3.4. Antimicrobial Resistance Profile of the Game Origin Pathogenic Strains

AMR has been identified by the World Health Organization (WHO) as one of the top ten global public health threats confronting humanity. Without intervention, it is estimated that global deaths attributable to AMR could reach 10 million annually by 2050 [79]. The administration of antibiotics is a cornerstone of contemporary medicine; however, the rise of AMR, particularly within the *Enterobacteriaceae*, is escalating into a global crisis [8]. The emergence of strains resistant to most, if not all, available antimicrobials presents significant challenges to public health [7].

Most AMR observed in *Enterobacteriaceae* arises from acquiring mobile genetic elements, such as plasmids, which facilitate horizontal gene transfer across species and even genera boundaries. Mutations in chromosomal genes also contribute significantly, enhancing resistance to various antimicrobial classes, including both newly introduced and older antimicrobials that are being reconsidered for treating MDR organisms [80].

Wild animals, due to limited direct exposure to antimicrobials, were traditionally expected to exhibit low levels of AMR. *Enterobacteriaceae* bacteria isolated from wild ungulate meat exhibited the highest resistance to tetracycline (TET) and ampicillin (AMP), followed by resistance to amoxicillin–clavulanic acid (AMC), as reflected in Table 4.

However, increasing interactions with humans and livestock have significant impacts on their bacterial flora [28]. Dias et al. [28] assessed AMR in *E. coli* and *Salmonella* isolated from wild ungulates, testing multiple antimicrobials. *E. coli* isolates exhibited the highest resistance to ampicillin (9.87%), followed by tetracycline (8.55%), streptomycin (4.61%), and co-trimoxazole (3.95%). MDR was detected in 3.3% of isolates, predominantly from wild boar and red deer. Conversely, a *Salmonella* strain from a wild boar sample demonstrated susceptibility to all tested antimicrobials.

In Norway, Lillehaug et al. [16] reported AMR in only 3 out of 137 *E. coli* strains (2.2%) from moose, red deer, and roe deer. Among reindeer isolates, three exhibited MDR, with streptomycin resistance being the most frequent. Costa et al. [81] found varied resistance levels in *E. coli* from wild animals in Portugal, with resistance to tetracycline, streptomycin, ampicillin, and trimethoprim–sulfamethoxazole ranging from 19% to 35%. Bertelloni et al. [61] observed high resistance levels in *E. coli* isolated from wild boars in Tuscany, particularly against β-lactam antimicrobials. Most isolates showed resistance to cephalothin, amoxicillin–clavulanic acid, and ampicillin (165/175, 152/175, and120/175), respectively, with lower resistance to enrofloxacin and gentamicin (24/175), while minimal resistance was noted for trimethoprim–sulfamethoxazole (3/175) and chloramphenicol (1/175). Even if these percentages seem uncommon and worrying, in Italy, different authors detected similar levels of resistance among *Enterobacteriaceae* from wild animals other than wild boar [82,83,84,85].

**Table 4 pathogens-13-01046-t004:** Antimicrobial Resistance (AMR) of *Enterobacteriaceae* from wild ungulates.

Country	Wildlife Species	Sample Type	Method Used	Pathogens	Tested Antimicrobials (%)		References
					Resistance	Susceptibility	
**Portugal**	red deer roe deer wild boar	fecal	disk diffusion assay	*E. coli*	AMP (22), AMC (16), FOX (1), STR (7), SXT (9), TET (11)	CTX, CAZ, ATM, IPM, AMK, NA, CIP, CHL	[28]
**Norway**	moose red deer roe deer reindeer	fecal	microdilution technique	*E. coli*	OXY (7), STR (21), SMX (9), TMP (0.7)	AMC, AMP, CEF, CHL, ENR, FFC, GEN, NA, NEO	[16]
**Portugal**	deer wild boar	fecal	disk diffusion assay	*E. coli*	AMP (22), AMC (7), CTX (1.8), ATM (1.8), CAZ (0.9), GEN (6), TOB (4.5), STR (22), TET (35), SXT (19), NA (14), CIP (9), CHL (6),	N.A.	[81]
**Italy**	wild boar	fecal	disk diffusion assay	*E. coli*	AMP (69), AMC (87), FOX (30), KF (94), CTX (27), CHL (0.6), TET (45), SXT (1.7), ENR (14), GEN (14), STR (20), ATM (21)	IPM	[61]
**Poland**	red deer roe deer fallow deer wild boar European bison	fecal	microdilution technique	*E. coli*	AMP (2.5), NA (2), CIP (5), CHL (1), STR (6), KAN (1), GEN (3), SMX (23), TMP (5), TET (5), COL (1)	CTX, CAZ, FLR	[59]
**Italy**	wild boar	fecal MLN	microdilution technique	*E. coli*	AMC, AMP, PIP, CTX, CAZ, FEP (100); SXT (75); CLO (37.5); CIP (62); LEV, TOB (31); GEN (37), AMC (19)	AMK, ERT, IPM, MER, TGC, FOS, TZP, COL	[86]
**Scotland**	red deer roe deer fallow deer sika deer	fecal	microdilution technique	*E. coli*	CPO (6.5) CIP (0.3), TET (22)	MER	[87]
**Italy**	wild boar	Liver	disk diffusion assay	*Salmonella*	AMP (9.6), AMC (6), CHL (0.4), KF (6), CTX (1), COL (0.4), CAZ (0.8), ENR (0.5), GEN (3), KAN (6), NA (0.8), STR (11), SSS (96), SXT (22), TET (20)	CIP	[39]
**Italy**	wild boar	fecal carcass MLN	disk diffusion assay	*Salmonella*	-	AMK, AMP, AMC, AZM, CZS, FOX, CRO, CTX, CAZ, CIP, DOX, IPM, KAN, LEV, MER, NA, STR, S, TET, SXT	[40]
				*Yersinia*	AMC + AMP + FOX (42), AMC + AMP (49), AMP (4)	N.A.	
**Italy**	wild boar	liver	disk diffusion assay	*Yersinia*	GEN, KAN (2); STR (1); STX (3); TET (0.8)	CHL, ENR	[72]
**Italy**	wild boar	serum fecal	disk diffusion assay	*Salmonella*	AMP (3.7), AMC (1.8), CZO (1.8), CEF (4), CTX (1.8), STR (18.5), GEN (5.5), SMX (15), NA (1.8), TET (5.5), COL (15)	KAN, NEO, ENR, CHL	[35]
**Italy**	wild boar	fecal liver spleen	disk diffusion assay	*Salmonella*	STR (55.6), KF (11), IPM (5.56), TET + ENR + NIT + NA + STR (6.67)	N.A.	[38]

AMC—Amoxicillin/Clavulanate, AMK—Amikacin, AMP—Ampicillin ATM—Aztreonam, AZM—Azithromycin, CAZ—Ceftazidime, CEF—Ceftiofur, CHL—Chloramphenicol, CIP—Ciprofloxacin, COL—Colistin, CPO—Cefoperazone, CRO—Ceftriaxone, CTX—Cefotaxime, CZO—Cefazoline, CZS—Cefazolin, DOX—Doxycycline, ENR—Enrofloxacin, ERT—Ertapenem, FEP—CEFEPIME, FFC—Florfenicol, FLR—Florfenicol, FOS—Fosfomycin, FOX—Cefoxitin, GEN—Gentamicin, IPM—Imipenem, KAN—Kanamycin, KF—Cephalothin, LEV—Levofloxacin, MER—Meropenem, NA—Acid Nalidixic, NIT—Nitrofurantoin, NEO—Neomycin, OXY—Oxitetraciclină, PIP—Piperacillin, S—Sulphonamide, STR—Streptomycin, SMX—Sulfamethoxazole, SSS—Triple Sulpha, SXT—Sulfamethoxazole/Trimethoprim, TET—Tetracycline, TGC—Tigecycline, TMP—Trimethoprim, TOB—Tobramycin, TZP—Piperacillin/Tazobactam, N.A.—not available.

In Poland, Wasyl et al. [59] detected resistance in E. coli from feces (n = 660) of wild ungulates (red, roe, fallow deer, European bison, and wild boar), with the highest resistance to sulfamethoxazole, streptomycin, ampicillin, trimethoprim, and tetracycline (1.3–6.6%). No significant differences were observed between boar and ruminant isolates. Most deer and bison isolates showed no resistance.

In Serbia, Velhner et al. [54] detected MDR phenotypes in seven isolates, each exhibiting distinct genomic macrorestriction profiles. PCR analysis and sequencing revealed diverse resistance genes, gene cassettes, and cassette arrays in these MDR isolates. Fluoroquinolone resistance was observed in five *E. coli* isolates, specifically in two from roe deer, one from deer, and two from wild boar. Mercato et al. [86] comprehensively evaluated antimicrobial susceptibility profiles for 16 ESBL-producing *E. coli* strains. The study revealed 100% non-susceptibility to penicillins, 3rd-generation cephalosporins (3GCs), 4th-generation cephalosporins (4GCs), tetracyclines, and monobactams. Resistance rates were 75% to trimethoprim–sulfamethoxazole, 37.5% to chloramphenicol, 62.5% to ciprofloxacin, and 31.25% to levofloxacin. Among aminoglycosides, resistance was observed at 31.25% to tobramycin, 37.5% to gentamicin, and 18.75% to amoxicillin/clavulanate. All isolates were fully susceptible to carbapenems, amikacin, tigecycline, fosfomycin, piperacillin/tazobactam, and colistin. Notably, all suspected ESBL-producing *E. coli* exhibited an MDR profile, with a significant proportion (greater than 60%) showing non-susceptibility to fluoroquinolones. This finding is of particular concern due to the frequent clinical use of fluoroquinolones for treating infections. Elsby et al. [87] found significant resistance to tetracycline and cefpodoxime in AMR *E. coli* from deer fecal samples in Scotland, although no resistance to meropenem was detected.

Razzuoli et al. [39] investigated the prevalence of AMR *Salmonella* spp. strains within the wild boar population in Liguria, Italy. Of the 260 strains analyzed, 94.6% (246/260) exhibited resistance to at least one of the tested antimicrobials. Specifically, 40% (98/260) were resistant to two or more antimicrobials, 17.3% (45/260) to three or more, and 9.6% (25/260) to four or more antimicrobials. The highest resistance rates were observed against a combination of sulfadiazine, sulfamerazine, and sulfamethazine, with 96% of the strains demonstrating resistance to these compounds. Conversely, less than 1% of the strains were resistant to chloramphenicol, colistin, ceftazidime, enrofloxacin, and nalidixic acid, and no strains showed resistance to ciprofloxacin. Intermediate susceptibility was most commonly observed for kanamycin (43%), streptomycin (30.2%), and tetracycline (23.4%). The observed AMR to these molecules is lower than that reported in other studies conducted in wild boars; however, these studies considered a lower number of *Salmonella* spp. strains [35,38,88].

Siddi et al. [40] reported that all *Salmonella* isolates (three/three, 100%) were susceptible to all tested antimicrobials. Regarding *Y. enterocolitica* isolates, three different AMR profiles were identified: 10/24 (41.7%) showed resistance to amoxicillin–clavulanic acid, ampicillin, and cefoxitin (AmcAmpFox); 11/24 (48.8%) showed resistance to amoxicillin–clavulanic acid and ampicillin (AmcAmp); 1/24 (4.2%) showed resistance to ampicillin (Amp); and 2/24 (8.3%) were sensible to all antimicrobials tested. Overall, 22/24 (91.7%) of *Y. enterocolitica* isolates showed phenotypic resistance to at least one beta-lactam compound. Comparably, Modesto et al. [72] reported that 61.9% (n = 78) exhibited resistance to at least one antimicrobial among the *Yersinia* isolates tested. Specifically, 85.71% of the isolates were resistant to ampicillin, 23.8% to Triple-Sulfa and sulfisoxazole, and 7.14% to ceftiofur. Resistance to chloramphenicol and enrofloxacin was not detected, and the strains demonstrated very low resistance to streptomycin and tetracycline (0.79%); an increasing trend in resistance was observed for ampicillin, Triple-Sulfa, sulfisoxazole, and ceftiofur. Additionally, regarding MDR, twelve strains were resistant to two antimicrobials, fourteen to three antimicrobials, five to four antimicrobials, and nine to five antimicrobials.

The data indicate a consistent pattern of rising AMR in wild ungulates across Europe. Although these animals are less exposed to antibiotics, they are increasingly showing resistance due to their interactions with human activities. This trend underscores the critical need for ongoing surveillance and robust stewardship of antimicrobial use. Effective monitoring should encompass both domestic and wild animal populations to address the spread of resistance comprehensively. Developing novel therapeutic strategies and alternative approaches to antimicrobial use is essential to counteract the escalating threat of AMR.

The rising prevalence of AMR in wild ungulates highlights the interconnectedness of ecosystems and the importance of a One Health approach to managing AMR. This approach recognizes that human, animal, and environmental health are intrinsically linked and must be addressed together to mitigate the global threat of AMR.

## 4. Conclusions

The analysis of game meat production and the prevalence of foodborne pathogens in wild ungulates across Europe reveals significant public health implications. While game meat is a traditional and valued part of European diets, it presents unique challenges due to its association with various pathogens and AMR bacteria. However, the economic impact of the summarized results cannot be accurately estimated because game meat consumption occupies different weights in the consumer’s diet within European countries. Furthermore, the possible negative impact on human and domesticated animals’ health cannot be neglected because wildlife has started to adapt to semi-urban or even urban areas, constituting an important reservoir for the persistence of drug-resistant pathogenic strains within the environment.

The reviewed studies highlight that wild boars are major reservoirs for *Salmonella* spp., *E. coli*, and *Yersinia* spp., with considerable variability in pathogen prevalence across different species and regions. Specifically, wild boars and other ungulates like deer are frequently implicated in the environmental dissemination of pathogenic *E. coli* and *Yersinia*. The rising levels of AMR in these wildlife populations, despite limited direct exposure to antimicrobials, is a growing concern. This resistance is likely linked to human activities and environmental contamination.

To mitigate these risks, it is crucial to implement stringent monitoring and surveillance programs for both game meat safety and AMR. Emphasizing a One Health approach, which integrates human, animal, and environmental health strategies, will be essential in addressing these issues comprehensively. Future research should focus on the detailed characterization of pathogens and resistance patterns in wildlife to develop effective public health interventions and ensure the safety of game meat.

## Figures and Tables

**Figure 1 pathogens-13-01046-f001:**
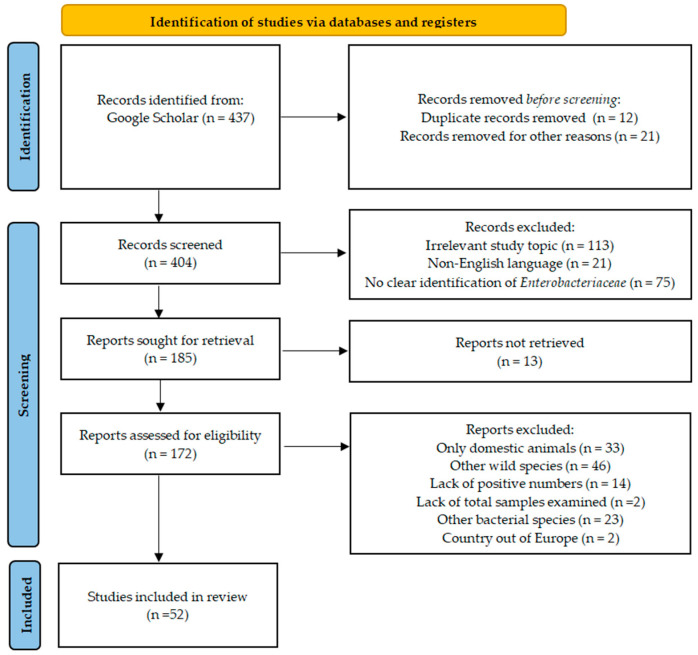
Identification of studies regarding the prevalence and AMR of *Enterobacteriaceae* members in wild ungulates via databases using PRISMA guidelines.

**Figure 2 pathogens-13-01046-f002:**
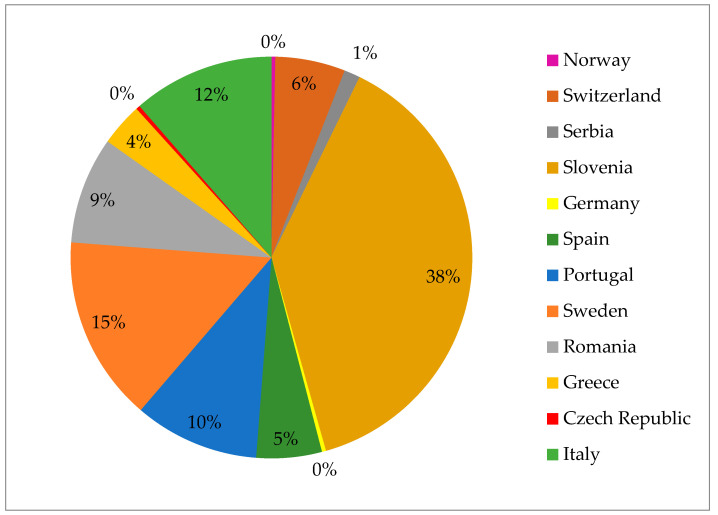
Overall prevalence of *Salmonella* spp. in game animals, recorded by country.

**Figure 3 pathogens-13-01046-f003:**
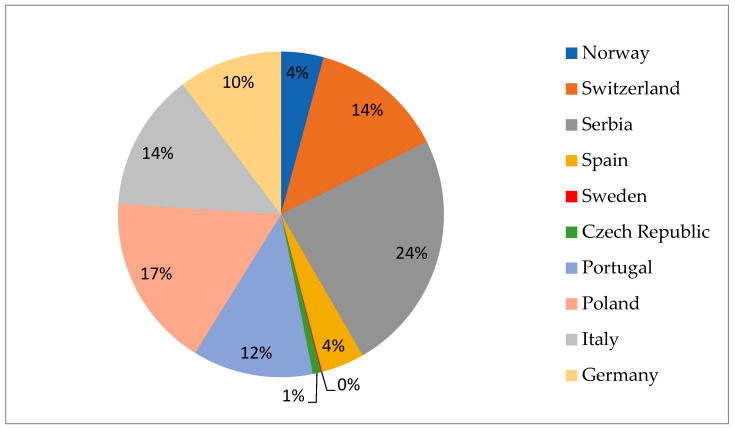
Overall prevalence of *E. coli* in game animals, recorded by country.

**Figure 4 pathogens-13-01046-f004:**
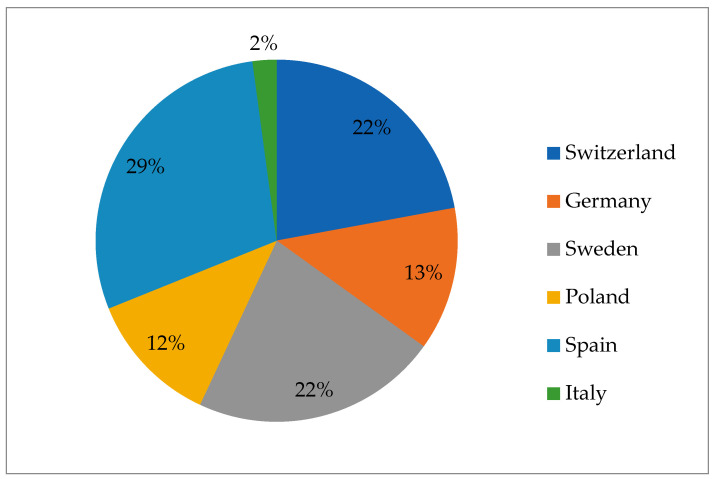
Overall prevalence of *Yersinia* spp. in game animals per country.

**Table 1 pathogens-13-01046-t001:** Prevalence of *Salmonella* spp. in game meat published by studies conducted within mainland Europe.

Country	Sample Type	Detection Modalities	Investigation Period	Wildlife Species Examined	No. of Positive (%)	No. of Examined	Serotypes (No. of Isolated Strains)	References
	Non-European Union countries							
**Norway**	fecal	S.M.M.	2001–2003	roe deer red deer moose reindeer	0	611	-	[16]
**Switzerland**	tonsils	N.A.B.M.	2007–2008	wild boar	19 (12.4)	153	*S. enteritidis* (6) *S. veneziana* (1) *S. stourbridge* (1)	[17]
	fecal	S.M.M.	2011	red deer roe deer chamois ibex	0	293	-	[18]
	diaphragm	I.B.M.	2020	wild boar	21 (17)	126	N.A.	[19]
**Serbia**	fecal MLNs	N.A.B.M.	2013–2024	wild boar	14 (1.6)	850	*S. enteritidis* (9) *S. typhimurium* (4) *S. infantis* (1)	[20]
	European Union countries							
**Slovenia**	serum	I.B.M.	2003–2004	wild boar	85 (47.7)	178	N.A.	[21]
**Germany**	pooled meat	S.M.M.	2006–2007	wild boar roe deer red deer	0	289	N.A.	[22]
**Spain**	serum	I.B.M.	2004–2007	wild boar	30 (11.3)	265	N.A.	[23]
	fecal carcass	S.M.M.	2009–2011	wild boar red deer fallow deer mouflon	5 (0.8)	637	N.A.	[24]
	fecal SLNs tonsils	N.A.B.M. S.M.M.	2010–2015	wild boar	86 (5.9)	1467	*S. enterica* (35) *S. diarizonae* (34) *S. salamae* (16) *S. houtenae* (1)	[25]
	serum	I.B.M.	2015–2019	wild boar	52 (19.3)	269	N.A.	[26]
**Portugal**	fecal	S.M.M.	2005–2006	wild boar	17 (22.1)	77	*S. typhimurium* (11) *S. rissen* (6)	[27]
	fecal	N.A.B.M.	2013–2014	wild boar red deer roe deer	1 (1.5)	67	N.A.	[28]
**Sweden**	fecal tonsils ILNs	N.A.B.M.	2010–2011	wild boar	9 (10.2)	88	*S. enterica* (4) *S. diarizonae* (2)	[29]
	fecal tonsils MLNs	N.A.B.M.	2014–2016	wild boar	24 (26.7)	90	N.A.	[30]
**Romania**	meat	S.M.M.	NA	wild boar	3 (10.7)	28	N.A.	[31]
**Greece**	serum	I.B.M	2006–2010	wild boar	4 (4.3)	94	N.A.	[32]
**Czech Republic**	fecal	S.M.M.	2014–2016	wild boar	1 (0.4)	242	N.A.	[33]
**Italy**	serum	I.B.M.	2005–2006	wild boar	105 (30.7)	342	N.A.	[34]
	serum fecal	S.M.M.	2010–2012	wild boar	309 (35)	882	*S. salamae* (13) *S. diarizonae* (7) *S. houtenae* (6) *S. fischerhuette* (4) *S. veneziana* (3) *S. napoli* (3) *S. kottbus* (3) *S. thompson* (3) *S. arizonae* (2) *S. toulon* (2) *S. burgas* (1) *S. tennelhone* (1) *S. ferruch* (1) *S. choleraesuis* (1) *S. paratyphi* (1) *S. stanleyville* (1) *S. typhimurium* (1) *S. enterica* (1)	[35]
	fecal MLNs	B.B.M.	2002–2010	wild boar red deer	5 (7.2)	69	*S. thompson* (2) *S. galiema* (1) *S. infantis* (1) *S. kottbus* (1)	[36]
	carcass MLNs	S.M.M.	2020	wild boar	6 (6.1)	98	*S. agama* (2) *S. enteritidis* (2) *S. zaiman* (1) *S. diarizonae* (1)	[37]
	liver spleen rectal swab	N.A.B.M.	2018–2020	wild boar	12 (4.2)	287	*S. london* (1), *S. infantis* (1), *S. rubislaw* (1), *S. newport* (2), *S. kottbus* (2), *S.* 1,40:z4,z23 (4), *S.* 50:r:1,5,7 (6)	[38]
	liver	N.A.B.M.	2013–2017	wild boar	540 (12.5)	4335	*S. enterica* (5) *S*. *enteritidis* (20) *S. typhimurium* (10) *S. typhimurium* monophasic variant 1,4,[5], 12:i:- (4) *S. infantis* (3) *S*. *newport* (8) *S. napoli* (21) *S. coeln* (9) *S. brandenburg* (3) *S. veneziana* (10) *S. thompson* (8) *S. canada* (8) *S. oxford* (7) *S. muenster* (6) *S. kottbus* (5) *S. gali* (4) *S. kimuenza* (6) *S. bnajul* (4) *S. stourbridge* (3) *S. juba* (2) *S. arechavaleta* (2) *S. atakpame* (1) *S. stoneferry* (1) *S. umbilo* (1) *S. goldocoast* (1) *S. grampiam* (1) *S. ablogame* (1) *S. massakory* (1) *S. bispebjerg* (1) *S. bahrenfeld* (1) *S. salamae* (53) S. *arizonae* (13), *S. diarizonae* (29) *S. houtenae* (7) *S. indica* (1)	[39]
	fecal carcass MLNs	S.M.M.	2020–2022	wild boar	3 (4.5)	66	*S. salamae* (1) *S. elomrane* (1) *S. enterica* (1)	[40]
	liver muscles	S.M.M.	2020–2022	wild boar red deer roe deer chamois	0	246	-	[41]
	fecal carcass liver	S.M.M.	2018–2023	wild boar	9 (1.4)	658	*S. stanleyville* (6) *S. typhimurium* (3)	[6]

N.A.: not available. MLNs: mesenteric lymph node, ILNs: ileocecal lymph nodes, SLNs: submandibular lymph nodes, N.A.B.M.: nucleic acid-based methods. I.B.M.: immunological-based methods; B.B.M.: biosensor-based methods; S.M.M.: standardized microbiological methods (ISO).

**Table 2 pathogens-13-01046-t002:** Overview of the isolation frequency and pathogenic potential of *E. coli* in game meat reported by studies conducted in mainland Europe.

Country	Sample Type	Detection Modalities	Investigation Period	Species Examined	No. of Positive (%)	No. of Examined	Serotypes (No. of Isolated Strains)	References
	Non-European Union countries							
**Norway**	fecal	N.A.B.M.	2001–2003	roe deer red deer moose reindeer	104 (16.8)	618	O26 (5) O103 (85) O145 (14)	[16]
**Switzerland**	fecal	N.A.B.M.	2011	roe deer red deer chamois ibex	127 (53.1)	239	N.A.	[18]
**Serbia**	fecal	N.A.B.M.	2013–2014	wild boar roe deer red deer	100 (95.2)	105	N.A.	[54]
	European Union countries							
**Spain**	fecal	N.A.B.M.	2004–2005	roe deer roe deer fallow deer mouflon	58 (23.9)	243	N.A.	[55]
	fecal	N.A.B.M.	2007–2008	wild boar	19 (8.9)	212	O157:H7 (7) O157:H21 (2) O23:H21 (2) O6:H10 (1) O104/O127:H (1) O104/O127:H1/H12 (1) O109:H (1) O127:H2 (1) O142:H8/H21 (1) O146:H21 (1) ONT:H7 (1)	[51]
	fecal	N.A.B.M., S.M.M.	2009–2010	wild boar roe deer	116 (26.3)	441	O146:[H28] O146:[H21] O2:H6 O75:[H8] O103:H28	[52]
	fecal	N.A.B.M.	2009–2011	wild boar Iberian ibex	6 (2.2)	277	O157:H7 (6)	[56]
**Sweden**	fecal tonsils ILNs	N.A.B.M.	2010–2011	wild boar	0	319	-	[29]
**Czech** **Republic**	fecal	N.A.B.M.	2014–2016	wild boar	2 (3.3)	242	O157 (2)	[33]
**Portugal**	fecal	N.A.B.M.	2013–2014	wild boar roe deer red deer	64 (96)	67	N.A.	[28]
	fecal	N.A.B.M.	2017–2019	wild boar red deer	45 (28.7)	157		[57]
	fecal	N.A.B.M.	2018–2020	wild boar red deer fallow deer mouflon	83 (45.8)	181	N.A.	[58]
**Poland**	fecal	N.A.B.M.	2012–2014	wild boar roe deer red deer fallow deer bison	553 (83.7)	660	N.A.	[59]
	fecal	N.A.B.M.	2020–2021	roe deer red deer	35 (17.4)	201		[60]
**Italy**	fecal	N.A.B.M.	2018–2019	wild boar	175 (87.5)	200	N.A.	[61]
	fecal	N.A.B.M.	2016–2018	red deer	40 (19.9)	201	O146:H28 (10) O113:H4 (4) O187:H28 (4) O91:H14 (3) O27:H30 (3) ONT:H49 (3) O91:H21 (1) O104:H7 (1) O174:H8 (1) O178:H19 (1)	[62]
	fecal MLNs carcass	N.A.B.M. S.M.M.	2020–2022	wild boar	154 (85)	181	N.A.	[40]
	liver	N.A.B.M. S.M.M.	2020–2022	wild boar roe deer red deer chamois	0	100	N.A.	[41]
**Germany**	fecal	N.A.B.M.	2016–2017, 2019	wild boar roe deer	1278 (40.8)	3129	N.A.	[63]

N.A.: not available, MLNs: mesenteric lymph node, ILNs: ileocecal lymph nodes, N.A.B.M.: nucleic acid-based methods, S.M.M.: standardized microbiological methods (ISO).

**Table 3 pathogens-13-01046-t003:** Prevalence of *Yersinia* spp. in game meat reported in European studies over the past two decades.

Country	Sample Type	Detection Modalities	Investigation Period	Species Examined	No. of Positive (%)	No. of Examined	Species (No. of Isolates)	References
	Non-European Union countries							
**Switzerland**	fecal tonsils	N.A.B.M. B.B.M.	2007–2008	wild boar	68 (44.4)	153	*Y. enterocolitica* *Y. pseudotuberculosis*	[73]
	European-Union countries							
**Germany**	raw meat *	N.A.B.M.	2006	N.A.	23 (38.3)	60	*Y.enterocolitica*	[74]
	tonsils	N.A.B.M	2012–2013	wild boar	21 (18.9)	111	*Y. frederiksenii* (2) *Y. enterocolitica* (19)	[75]
**Sweden**	fecal tonsils ILNs	N.A.B.M.	2010–2011	wild boar	31 (35.2)	88	*Y. enterocolitica* *Y. pseudotuberculosis*	[29]
	fecal tonsils MLNs	N.A.B.M.	2014–2016	wild boar	48 (53.3)	90	*Y. enterocolitica* (28) *Y. pseudotuberculosis* (20)	[30]
**Poland**	rectal swab	N.AB.M., S.M.M.	2012–2013	wild boar	40 (26.5)	151	*Y. enterocolitica*	[76]
	tonsils peritoneum perineum	N.A.B.M. S.M.M.	2013	wild boar roe deer red deer	30 (53.5)	56	*Y. enterocolitica* (52)	[77]
	rectal swab	S.M.M.	2013–2014	wild boar roe deer red deer fallow deer	186 (21.7)	857	*Y. enterocolitica* (218)	[69]
**Spain**	serum tonsils	N.A.B.M.	2001–2012	wild boar	294 (58.2)	505	*Y. enterocolitica* *Y. pseudotuberculosis*	[71]
**Italy**	carcass	N.A.B.M. S.M.M.	2008–2010	wild boar roe deer red deer chamois	18 (7.2)	251	*Y. enterocolitica* (6)	[78]
	fecal spleen liver	S.M.M.	2018–2020	wild boar	71 (24.7)	287	*Y. enterocolitica* (54) *Y. frederiksenii*, *Y. intermedia* (17)	[38]
	carcass swabs MLNs	S.M.M.	2020	wild boar	3 (4.6)	64	*Y. frederiksenii,* (1) *Y. bercovieri* (1) *Y. aldovae* (1)	[37]
	liver	S.M.M.	2013–2018	wild boar	126 (2.5)	4890	*Y. enterocolitica*	[72]
	fecal carcass swabs MLNs	S.M.M.	2020–2022	wild boar	20 (30.3)	66	*Y. enterocolitica*	[40]
	liver	S.M.M.	2020–2022	wild boar roe deer red deer chamois	8 (7.9)	101	*Y. enterocolitica*	[41]

N.A.: not available. MLNs: mesenteric lymph node, ILNs: ileocecal lymph nodes. * The specific species from which the game meat originates is not mentioned, B.B.M.: biosensor-based methods, N.A.B.M.: nucleic acid-based methods, S.M.M.: standardized microbiological methods (ISO)

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
