# Peer review of "Systematic Review of the Occurrence and Antimicrobial Resistance Profile of Foodborne Pathogens from Enterobacteriaceae in Wild Ungulates Within the European Countries"

_pathogens, 2024, doi:10.3390/pathogens13121046_

Round 1
Reviewer 1 Report (Previous Reviewer 3)
Comments and Suggestions for Authors
Thr review responds well to the previous comments. It is well revised.
Author Response
Dear respected reviewer,
Thank you for your time and valuable suggestions during the review process, as well as appreciation to the address of our submission.
Yours faithfully,
dr. Imre
Reviewer 2 Report (Previous Reviewer 2)
Comments and Suggestions for Authors
Appreciating your efforts.
Best wishes,
Reviwer
Author Response
Dear respected reviewer,
Thank you for your time and valuable suggestions during the review process, as well as appreciation to the address of our submission.
Yours faithfully,
dr. Imre
Reviewer 3 Report (Previous Reviewer 1)
Comments and Suggestions for Authors
Authors have made effective revisions, and it is suggested to accept the manuscript.
Author Response
Dear respected reviewer,
Thank you for your time and valuable suggestions during the review process, as well as your appreciation for the address of our submission.
Yours faithfully,
dr. Imre
This manuscript is a resubmission of an earlier submission. The following is a list of the peer review reports and author responses from that submission.
Round 1
Reviewer 1 Report
Comments and Suggestions for Authors
This manuscript studies the epidemic distribution of Enterobacteriaceae from wild ungulates and their antimicrobial resistance through reference analysis, which is very meaningful. However, it should be noted that if the content described in this manuscript belongs to the category of review, authors are advised to follow the writing template of review, set several key titles, rather than the current template according to the section including materials and methods, results and discussion. If it is more inclined to analyze and study the relevant data in published literature, it is suggested to make statistical analysis of the data involved according to the template of research paper, and list relevant data charts instead of simply listing literature data. If authors choose the latter, they will get the scientific data of dominant pathogens and antimicrobial resistance, and the article will be more valuable.
In additon, ESβL-producing is generally written as ESBL-producing.
Author Response
#Reviewer1:
Manuscript ID: 3210883
This manuscript studies the epidemic distribution of Enterobacteriaceae from wild ungulates and their antimicrobial resistance through reference analysis, which is very meaningful.
Answer: Dear reviewer,
Thank you very much for your time to review our manuscript, and for your positive comments and appreciation! Below you can read our answers to the raised concerns.
However, it should be noted that if the content described in this manuscript belongs to the category of review, authors are advised to follow the writing template of review, set several key titles, rather than the current template according to the section including materials and methods, results and discussion.
Answer: Thank you for your valuable feedback regarding the manuscript structure. We would like to clarify that we adhered to the PRISMA checklist in the preparation of this systematic review. Given that our review examines 3 distinct pathogens, we opted to discuss the results and implications for each pathogen separately. This approach was chosen to enhance clarity and facilitate a better understanding of the findings presented in the manuscript.
If it is more inclined to analyze and study the relevant data in published literature, it is suggested to make statistical analysis of the data involved according to the template of research paper, and list relevant data charts instead of simply listing literature data. If authors choose the latter, they will get the scientific data of dominant pathogens and antimicrobial resistance, and the article will be more valuable.
Answer:
In our manuscript, we have primarily focused on a qualitative synthesis of the literature to provide a comprehensive overview of the dominant pathogens and their antimicrobial resistance profiles. According to the reviewer's recommendation, to improve the statistical overview data we included 3 charts that illustrate the overall prevalence of each targeted pathogen across different countries. These visual representations aim to provide a clearer understanding of the distribution of dominant pathogens.
In addition, ESβL-producing is generally written as ESBL-producing.
Answer: We have changed it to ESBL-producing, as it is the standard way to refer to it.
Thank you again!

Reviewer 2 Report
Comments and Suggestions for Authors
Dear Correspondance Author,
Appreciating your efforts. Please find hereunder and in the attached file my comments and suggestions that might enrich your manuscript.
General comments and suggestions:
1- Try to highlight also, the economical impact of your targetted review topics with the provided health impacts.
2- Provide a difinition stateent for "Game meat in the beginning of the abstract such as: "Game meat is meat derived from wild animals.".
3- Journal format:
a) for the reference (FACE, 2021) in line: 33 and which in Lines: 118, 540 and 704-705.
b) Unify the number format: letters of digits. (eg. Lines: 64 and 228)
c) abbreviations format. (eg. Line: 232 for STEC).
3- List of criteria for the excluded records in Figue 1.
4- Providing a chart per country for the wide range 0-47.7% (lines: 158-160).
5- In tables (1 and 2), try to ssplit the data by sample type.
6- Some statements required supportive reference. (eg. Lines: 57 and 210).
Best wishes,
Reviwer

Author Response
#Reviewer 2:
Manuscript ID: 3210883
Dear Correspondance Author,
Appreciating your efforts. Please find hereunder and in the attached file my comments and suggestions that might enrich your manuscript.
Answer: Dear reviewer,
Thank you very much for your time to review our manuscript, and for your positive comments and appreciation! Below you can read our answers to the raised concerns.
General comments and suggestions:
1- Try to highlight also, the economical impact of your targetted review topics with the provided health impacts.
Answer: In agreement with the reviewer's requirement, the following sentences were inserted in the revised version of the manuscript:
"In rural areas within mainland Europe, hunters are regarded as primary producers of game meat, with an important contribution to the development of local economies, supporting thus the sustainable meat production [1,2]."
"However, the economic impact of the summarized results cannot be accurately estimated, because game meat consumption occupies different weights in the consumer's diet within the European countries. Furthermore, the possible negative impact on human and domesticated animals’ health cannot be neglected because wildlife started to adapt to semi-urban or even urban areas constituting an important reservoir for the persistence of drug-resistant pathogenic strains within the environment."
2- Provide a difinition stateent for "Game meat in the beginning of the abstract such as: "Game meat is meat derived from wild animals."
Answer: Dear reviewer,
“Game meat is derived from non-domesticated, free-ranging wild animals.”
This definition emphasizes the natural, free-ranging origin of the animals and birds from which game meat is obtained.
3- Journal format:
- a) for the reference (FACE, 2021) in line: 33 and which in Lines: 118, 540 and 704-705.
Answer: We have adjusted all the requirements.
- b) Unify the number format: letters of digits. (eg. Lines: 64 and 228)
Answer: Dear reviewer,
To unify the number format in the document based on the two lines you provided:
- Change “eight” to “8” in "1406 hospitalizations, and eight deaths, in line 64.
- Change “Six” to “6*” in "Six principal pathotypes of E. coli, in line 228.
This ensures a consistent use of digits throughout the text for numbers.
- c) abbreviations format. (eg. Line: 232 for STEC).
Answer: Thank you for the observation. We have adjusted the error. STEC stands for Shiga toxin-producing Escherichia coli.
3- List of criteria for the excluded records in Figue 1.
Answer: The exclusion criteria have been incorporated into Figure 1.
4- Providing a chart per country for the wide range 0-47.7% (lines: 158-160).
Answer: Dear reviewer,
The requested chart illustrating the prevalence range of 0-47.7% for Salmonella spp. in game animals, broken down by country, has been completed.
5- In tables (1 and 2), try to split the data by sample type.
Answer: With respect to the reviewer request, the authors would like to keep the original version of the tables. The included references in the present review presented the processed matrices within an extremely different way mentioning or no the correspondent total number of samples. Therefore, some articles provided only the total number of samples processed. For consistency, we have chosen to analyze the data using the total number of samples for each article. For a more comprehensive version of the table, the current version reflects the most complete presentation of the data.
6- Some statements required supportive reference. (eg. Lines: 57 and 210).
Answer: The references have been incorporated into the text.
Best wishes,
Reviwer
Thank you again!

Reviewer 3 Report
Comments and Suggestions for Authors
The manuscript is important in many areas of application. The review covered critical issues in food safety and antibiotic resistance with reference Wild Ungulates. The systemic review generated and summarized the data in a readable fashion. It would have been helpful if the tables include the detection modalities used in each listed reference.
The AMR is mentioned in the tile and sufficiently mentioned in the review. It will make the review more descriptive if there was a table generated for the AMR results with the resistance level and the protocol(s) used in the listed experiment.
With Regards,
Author Response
#Reviewer 3:
Manuscript ID: 3210883
The manuscript is important in many areas of application. The review covered critical issues in food safety and antibiotic resistance with reference Wild Ungulates. The systemic review generated and summarized the data in a readable fashion.
Answer: Dear reviewer,
Thank you very much for your time to review our manuscript, and for your positive comments and appreciations! Below you can read our answers to the raised concerns.
It would have been helpful if the tables include the detection modalities used in each listed reference.
Answer: We have complied with the proposed requirement and incorporated a new column in the tables, detailing the detection modalities used in each listed reference.
The AMR is mentioned in the tile and sufficiently mentioned in the review. It will make the review more descriptive if there was a table generated for the AMR results with the resistance level and the protocol(s) used in the listed experiment.
Answer: The authors completely agree the reviewer suggestion, and acknowledge the fact that during the preparation of the original version of the manuscript, they tried to conceive uniformly the structure of each chapter of the manuscript, meaning one suggestive table for each subheading concept. However, in case of “3.4. Antimicrobial resistance profile of the game origin pathogenic strains” subheading, the diversity of the recorded variables presented in each study limited us to generate a complete table with uniformly presented representative data for each study. The incomplete table that we tried to generate has the following shape:
No. |
Isolation source |
Species, serotype |
Resistance (%) |
References |
1. |
roe dee wild boar red deer |
E. coli |
Sulfametaxazole (23%), AMP (7%), TMP, CIP, TET (5%), STR (6%), GEN (3%), chloramphenicol, KAN (1%), HAL (2%) |
Wasyl et al., 2018 |
STR (4%), SMX (13%), TMP, TET, COL (2%) |
||||
2. |
ungulates |
E. coli |
AMP (3.3%), TET (7.78%), STR (3.3%), SXT (4%), AMC, Cefoxitin, colistin (1%) |
Dias et al., 2015 |
wild boar |
AMP (21%), TET (10%), CO-TRIM (9%), STR (7%) |
|||
3. |
ungulates |
E. coli |
OXY (7.1%), STR (21%), SXT (9.5%), TMP (0.7%) |
Velhner et al., 2018 |
4. |
ungulates |
Enterococcus faecium |
Bacitracin (25%), FLA (100%), OXY (25%) |
Lillehaug et al., 2005 |
Enterococcus faecalis |
ERY, OXY, STR (6.7%), VIRGIN (100%) |
|||
5. |
ungulates |
Yersinia spp. |
AMP (85%), Triple-sulfa, Sulfisoxasole (23%), ceftiofur (7%) |
Modesto et al., 2021 |
6. |
ungulates |
Yersinia spp. |
AMC, AMP, CEF (41%) |
Siddi et al., 2023 |
7. |
E. coli |
AMP (3%), AMC (7.1%), CEF (1.8%), STR (22%), TET (34%), STX (18%), NAL (14%), CIP (8.9%), clorthrimazol (6.3%), GEN (6.3%), TOB (4.5%) |
Costa et al., 2008 |
|
8. |
wild boar |
E. coli |
Cefalotin (94%), AMC (86%), AMP (68%), TET (44%), FOX (29%), ENR, GEN (13%), STR, AZT (20%) |
Bertelloni et al., 2020 |
9. |
S. Typhimurium S. Brondenburg S. enterica sub. Salamae |
SSS (96%), TET (20%), STX (21%), STR (10%), KAN (5.7%), Cefalotin (6.2), AMP (9.6%) |
Razzuoli et al., 2021 |
|
10. |
S. enterica sub. Salamae S. enterica sub. Diarizonae S. enterica sub. Hautenae |
Sulfonamides (92%), SXT (14%), COL (14), STR (18%), GEN, TET (5.5%), CEF, AMP (3.7%), cefazolin, cefotaxime, AMC (1.8%) |
Zottola 2012 |
Taking these into consideration, the research team would like to present the antimicrobial resistance profile of the game origin pathogenic strains in the present form.
Special thanks for your understanding!
